# Cross-Modality Person Re-Identification Algorithm Based on Two-Branch Network

**Jianfeng Song \*** , **Jin Yang, Chenyang Zhang and Kun Xie**

School of Computer Science and Technology, Xidian University, Xi'an 710071, China
* Correspondence: jfsong@mail.xidian.edu.cn

**Abstract:** Person re-identification is the technique of identifying the same person in different camera shots, known as ReID for short. Most existing models focus on single-modality person re-identification involving only visible images. However, the visible modality is not suitable for low-light environments or at night, when crime is frequent. In contrast, infrared images can reflect the nighttime environment, and most surveillance systems are equipped with dual-mode cameras that can automatically switch between visible and infrared modalities based on light conditions. In contrast to visible-light cameras, infrared (IR) cameras can still capture enough information from the scene in those dark environments. Therefore, the problem of visible-infrared cross-modality person re-identification (VI-ReID) is proposed. To improve the identification rate of cross-modality person re-identification, a cross-modality person re-identification method based on a two-branch network is proposed. Firstly, we use infrared image colorization technology to convert infrared images into color images to reduce the differences between modalities and propose a visible-infrared cross-modality person re-identification algorithm based on Two-Branch Network with Double Constraints (VI-TBNDC), which consists of two main components: a two-branch network for feature extraction and a double-constrained identity loss for feature learning. The two-branch network extracts the features of both data sets separately, and the double-constrained identity loss ensures that the learned feature representations are discriminative enough to distinguish different people from two different patterns. The effectiveness of the proposed method is verified by extensive experimental analysis, and the method achieves good recognition accuracy on the visible-infrared image person re-identification standard dataset SYSU-MM01.

**Keywords:** person re-identification; cross-modality; modal transformation; two-branch neural network

## 1. Introduction

With the widespread use of video surveillance systems in cities, techniques that use images captured by cameras to determine whether pedestrians appearing in different images are the same person and predict their behavior from the generated trajectories have been widely used in the fields of smart video surveillance and criminal investigations. The technique, which uses computer vision and machine learning to retrieve the presence of a person with the same identity in images or video sequences, is known as person re-identification (Re-ID). Person re-identification has not only a very pressing application need but also a very important research value. In recent years, person re-identification has received widespread attention from academia and industry and is a research hotspot in computer vision.

Traditional visible-light surveillance cameras are no longer suitable for scenes that require 24-h surveillance, such as road traffic and prisons. At present, many surveillance systems and intelligent devices are equipped with automatic visible-to-infrared mode switching functions. The problem of person re-identification in visible and infrared modalities needs to be solved urgently. The main task of cross-modality person re-identification is to retrieve a visible image or an infrared image of a specific individual, given the image library in both modalities, to match images belonging to the same individual. It plays an

important role in various surveillance videos and intelligent applications and has attracted increased attention in research sessions.

At present, we have achieved good results in the field of single-modality person re-identification, such as person re-identification based on visible light mode, which has reached 95% rank-1 under the Market1501 dataset [1,2], but there are still big challenges in the field of cross-modality person re-identification. There are significant differences between the images captured in the two modalities: RGB images have three channels containing the visible light color information of red, green, and blue, while IR images have only one channel containing the intensity information of near-infrared light, and the wavelength ranges of the two are also different from the perspective of imaging principles. Different sharpness and lighting conditions can produce very different effects on the two types of images. Such inconsistent information distribution can disrupt the feature learning process of neural networks. Therefore, cross-modality person re-identification research focuses on reducing the negative impact caused by the differences between modalities.

To solve the above-mentioned problem, this study proposes a two-branch cross-modality person re-identification algorithm, which first uses infrared image colorization technology to preprocess infrared images and convert infrared data into color data to reduce the heterogeneous gap between inter-modality images to some extent, then inputs the two kinds of data into parameter-independent feature extraction networks separately, and then embeds the extracted cross-modality features into the shared network after dual constraint modeling of intra-modality and inter-modality features, allowing the network to learn better.

The main contributions of this study are as follows:

- applying the infrared image colorization technique to the field of cross-modality person re-identification.
- extracting the features of the two modalities separately and mapping the two modalities features into the same feature space.
- handling both intra-modality and cross-modality disparities simultaneously.

## 2. Related Works

### 2.1. Colorization of Infrared Images

In recent years, the rapid development and wide application of deep learning have provided a somewhat significant reference value for the image coloring field. Combining deep learning and infrared image coloring together reduces manual operations and interventions by putting a large number of infrared images into neural networks for training and learning, extracting image features, and thus achieving colorization of infrared images. To date, infrared image colorization remains a challenging topic with significant and far-reaching implications for this application area. Recently, convolutional neural network (CNN)-based methods have emerged as the predominant paradigm for nearly all computer vision tasks. CNNs have shown excellent performance in stereo vision [3], image classification [4], cross-spectral domain correlation, etc.

In 2016, Larsson [5] and Zhang [6] quantized the chromaticity space into discrete colors and performed logistic regression to predict the color histogram to handle multimodality. Larsson et al. processed grayscale images with VGG-16 and used spatially localized multilayer slices (supercolumns) as descriptors for each pixel. The system was de-trained from end to end and used to predict the chromaticity distribution and hue of a given supercolumn descriptor pixel. In the same year, Limmer et al. proposed an integrated method of deep learning techniques for the spectral transmission of NIR to RGB images [7]. The method utilizes a deep multiscale convolutional neural network (CNN) for direct estimation of low-frequency RGB values.

In 2017, Varga et al. used two parallel convolutional neural network architectures with the same structure and a reference image to perform colorization [8]. One of the CNNs uses the reference image to help the other CNN perform color prediction on the input image. Deshpande et al. implemented a low-dimensional embedding of the color domain using a variational self-encoder (VAE) to construct the loss term of the VAE decoder and avoid blurring of the output [9]. Finally, a conditional model of multimodality distribution of

grayscale images with color embedding was developed, which produces different colors for the samples of the conditional model.

In 2018, Suárez et al. [10] added a feature hierarchy to each layer in a stacked GAN architecture and used a polynomial loss function composed of intensity loss (MSE), structure loss (SSIM), and adversarial loss. The model can generate high-quality color infrared images. Dong et al. [11] proposed a method that first uses an encoder-decoder to convert NIR images into RGB images and then uses an assistant network to enhance edges and stabilize color regions.

In 2019, Mehri et al. [12] proposed a model of three-channel feedback, which solved the problem of being unable to obtain real values during the learning phase of unpaired datasets. For the reason that RGB images have more information than NIR images, Sun et al. [13] constructed an asymmetric model that considers different network capacities according to different conversion directions based on cyclic GAN. This model can deal with the problem of data unregistration caused by the brightness difference between RGB and NIR. Dong et al. [14] first used an edge-aware auto encoder decoder composed of an auto encoder decoder (AED) and an auxiliary assistant to generate RGB output. Furthermore, under the guidance of the weight map, the detailed information of the input NIR image was multi-resolution fused with the generated RGB image, so that more details were retained in the coloring result.

In 2020, Wang et al. [15] combined semantic segmentation and transfer learning in infrared colorization to obtain color images that are better in peak signal-to-noise ratio (PSNR) and structural similarity (SSIM). Among them, transfer learning can obtain rich color information in the case of insufficient training data, and semantic segmentation can be used as global prior information to make the boundary of the image clearer and significantly reduce the color error of the texture area. Sekiguchi et al. [16] combined a colorization network and a loss network and used perceptual loss and pixelwise loss functions, which can obtain a good structure with a limited number of NIR and RGB image pairs.

In 2021, Park et al. [17] exploited the correlation between individual NIR bands and RGB by using multi-band NIR images. It can successfully colorize the multi-band NIR images using a two-branch structure and the constraint of the proportional gradient between NIR and RGB. Kim et al. [18] proposed a near-infrared colorization model based on the correlation module of VAE and U-Net and enhanced texture and chrominance information by extracting the relationship between luminance and chrominance components.

In 2022, in order to improve the efficiency of infrared image colorization without affecting its performance, Jiang et al. [19] used 4×4 the Discrete Cosine Transform (DCT) to divide low-frequency and high-frequency details to reduce the difficulty of network training and the Residual (RIR) module to improve the colorization efficiency. Zhou et al. [20] introduced feature weights in UNet++ to obtain better minutiae coloring results. Furthermore, the brightness network was used to balance the overall color of the image so that the generated image is closer to the real image.

### 2.2. Cross-Modality Person Re-Identification

The main challenge facing cross-modality person re-identification is the huge differences between the two modalities. How to model the modality to better reduce the differences between the two modalities images and learn the robustness features shared between the two modalities is the key to current research. The two main research methods are representation-based learning and metric-based learning and then modality conversion-based learning methods have been proposed to achieve modality style conversion between RGB images and IR images, thus converting the cross-modality person re-identification problem into a person re-identification problem in a single modality.

In 2017, Wu et al. [21] proposed to define the cross-modality person re-identification problem for the first time in the field of person re-identification, analyzed three network architectures, and proposed a deep zero-complement data pre-processing method to compare and evaluate the performance of these four networks.

In 2018, Ye et al. [22] used a two-stream CNN network with identity loss and contrastive loss to learn multi-modality shareable features for cross-modal matching, and then solved the viewpoint variation problem of different cameras by compressing the features to the modalities. Using DCNN as the framework, Dai et al. [23] combined identification loss and cross-modality triplet loss, which minimizes the class of internal identification ambiguity and maximizes the class spacing of cross-modality similarity. Zhang et al. [24] learned the common features through the RGB branch and IR branch, which are represented by 3D tensors. Furthermore, a Contrastive correlation network (CCN) was used to capture the semantic differences between paired person images.

In 2019, Hao et al. [25] proposed a hyperspherical manifold embedding model. The modality transformation approach mainly uses the Generative Adversarial Network (GAN) as a modality transformer to transform human images from one modality to another and to achieve the interconversion of the two. Wang et al. [26] proposed a Dual-Level Discrepancy Reduction Learning (D2RL). Wang et al. [27] proposed an end-to-end alignment generation adversarial network for cross-modality person re-identification tasks. The pixel alignment module converts RGB images into IR images; the feature alignment module maps real and synthetic IR images into the same feature space and supervises the features using identity label-based classification and triadic loss; the joint discrimination module is responsible for discriminating between real and fake IR images, and the first two learn from each other through the identity consistency property to obtain robust features.

In 2020, Liu et al. [28] proposed a central triad loss, which calculates a mean value as the center for the features of each ID in a mini-batch and calculates the triad loss between class centers, which can relax the strict constraint between samples. Unlike previous GAN methods, Zhang et al. [29] proposed a TS-GAN approach, which guides a student model to extract discriminable features by pre-training a teacher model. First, a real RGB image was used to generate a fake IR image by GAN, and then the fake IR image and the real IR image were used as inputs to the teacher model to generate a feature mapping, which guides the student model in the backbone to generate a feature mapping as well, and the discriminative features can be obtained by extracting high-level semantic information in the high-level embedding layer. To better utilize the infrared image information, Fan et al. [30] proposed the Cross-Spectrum Image Generation (CSIG) method to generate images of multiple spectra and the Dual-Subspace Pairing Strategies (DSPS) to utilize the generated spectral images. The problem that the intra-class distance is larger than the inter-class distance in the infrared mode was solved. The Dynamic Hard Spectrum Mining-DHSM method was also proposed to optimize the randomness strategy for cross-spectrum generation to be more biased towards generating hard-trained samples.

In 2021, Fu et al. [31] systematically investigated the manually designed architecture and determined that proper separation of the batch normalization (BN) layers is the key to greatly facilitating cross-modality matching. Based on this observation, the best separation scheme was found for each BN layer. A new approach, called Cross-modality Neural Architecture Search (CMNAS), was proposed. It consists of a BN-oriented search space in which standard optimization can be accomplished by cross-modality tasks. Liu et al. [32] trained the grayscale maps generated from RGB maps together with IR maps to reduce the differences between modalities while also preserving the structural information of the original RGB maps; a new FC layer and BN layer were added in front of a single FC layer when calculating the ID loss, which can increase the efficiency of ID loss; three kinds of bidirectional losses were designed in ranking loss to compare comprehensively: cross-modality loss, intra-modality loss, and inter-modality loss, which also achieved better results.

In 2022, Hu et al. [33] reduced the modal gap between visible and infrared images and enhanced the feature representation by integrating DrRD, MiDR, and ROD to extract identity and domain-dependent features. Zhang et al. [34] proposed a comprehensive hybrid metric learning framework by combining four similarity constraints. The framework was compatible with any pairing-based loss function. Based on Circle loss [35], Liu [36] pro-

posed a new memory-enhanced one-way metric learning method for VI-ReID and solved the modal imbalance problem by creating memory banks based on two specific modalities.

## 3. Methodology

### 3.1. Design of Two-Branch Network

Since ResNet was proposed, it has been widely used in various fields of deep learning due to its simple and practical characteristics. Many algorithms in the VI-ReID domain are also completed using ResNet.

Two-branch networks were introduced as early as when Wu et al. [23] proposed the cross-modality person re-identification problem and are a common approach to accomplishing feature extraction in cross-domain tasks. The two-branch network structure first extracts single-modality features from the input RGB and IR images using two separate networks and then projects the extracted RGB and IR features into the shared feature space of VI-ReID using a parameter sharing network.

The data for the two modalities contains both modality-related feature information and identity-matching-related feature information. It is necessary to distinguish modality information from identity information and map more identity information to the shared feature space to improve the identity recognition capability under unified feature representation. Therefore, a two-branch network is designed for feature extraction of different modalities, in which the parameters of the shallow layer are independent to extract modality-specific information, which solves the problem of differences caused by modality between different data. Meanwhile, the two-branch network utilizes a partially shared structure to learn multimodal shareable features by extracting modality-specific information and modality-shared information simultaneously.

The two-branch cross-modality person re-identification network proposed in this paper uses ResNet50 as the backbone network, and its network structure is shown in Figure 1.

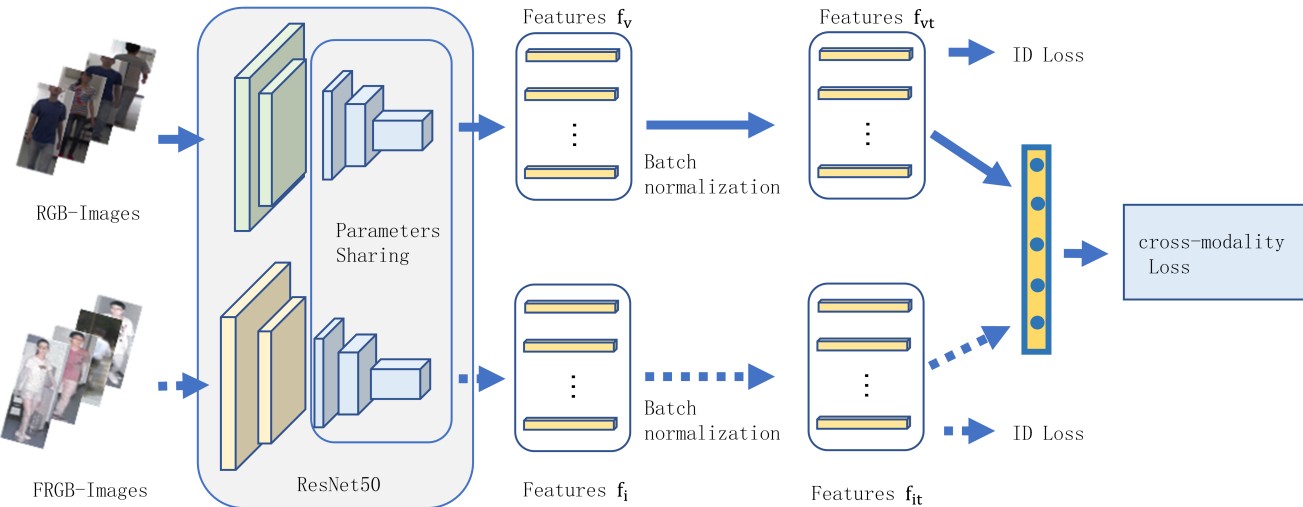

**Figure 1.** Two-branch network structure diagram.

According to the description in the literature [37], the backbone network ResNet50 can be divided into five parts, as shown in Table 1, which gives the naming of the ResNet50 network structure and the description of the corresponding layer parameters.

**Table 1.** ResNet50 neural network structure naming and parameter description.

| Name of Network Layer | Description of the Parameters |
|---|---|
| Conv1 | $7 \times 7, 64$, stride 2 <br> $3 \times 3$, max pool, stride 2 |
| Conv2 | $\begin{bmatrix} 1 \times 1, & 64 \\ 3 \times 3, & 64 \\ 1 \times 1, & 256 \end{bmatrix} \times 3$ |
| Conv3 | $\begin{bmatrix} 1 \times 1, & 128 \\ 3 \times 3, & 128 \\ 1 \times 1, & 512 \end{bmatrix} \times 4$ |
| Conv4 | $\begin{bmatrix} 1 \times 1, & 256 \\ 3 \times 3, & 256 \\ 1 \times 1, & 1024 \end{bmatrix} \times 6$ |
| Conv5 | $\begin{bmatrix} 1 \times 1, & 512 \\ 3 \times 3, & 512 \\ 1 \times 1, & 2048 \end{bmatrix} \times 3$ |

In the process of information extraction by the neural network, the shallow convolutional layer mainly captures the low-level visual information of the image, and the deeper the network is, the more high-level semantic information can be extracted, which contains more identity discriminative information. The data from both modalities in the cross-modality person re-identification task have shared identity feature information as well as modality-specific feature information, and the model needs to acquire more features related to identity information to achieve good results. Therefore, the designed two-branch network uses parameter-independent network structure in the shallow layer to extract the low-level visual information of the two modalities separately, and the parameter-sharing deep layer is used to extract the shared features.

The two-branch network structure has two main components: feature extraction and feature embedding. In the feature extraction part, the two branches input visible data and colored pseudo-visible data, respectively, which can capture modality-specific information about different images; the feature embedding part focuses on learning the shared space across modalities and characterizing the extracted features. The learning objectives mainly contain cross-modality and intra-modality constraints.

**Feature extraction.** The model uses ResNet50 as the backbone of the feature extraction network for the backbone network layers Conv1, Conv2, Conv3, Conv4, and Conv5, where Conv1 and Conv2 are shallow layers with no shared parameters, while Conv3, Conv4, and Conv5 are deep layers with shared network parameters, so that more features related to identity discrimination can be learned. As shown in Figure 1, given the visible light input data R or the infrared input data FR, the feature maps $f_v$ or $f_i$ are obtained through the backbone network ResNet50.

**Feature embedding.** To learn the discriminative information of two different modalities, we introduce a fully connected layer after the two-branch feature extractor, where the parameters of the fully connected layer are shared to model the shared modality information. Otherwise, the features learned by the two modalities may be in completely different subspaces. The shared structure serves as a projection function to project two different modalities into a common space. The feature map $f_v$ or $f_i$ map obtained in feature extraction is cut into n parts along the horizontal direction, and $n * 2048$ dimensional features are obtained using a global pooling layer on each part. In order to further reduce the feature dimension, a $1 \times 1$ kernel convolution layer and BatchNorm layer are used to reduce the dimension of each 2048-dimensional component feature, and finally a 256-dimensional feature expression is obtained. Therefore, for each input image R or FR, n

256-dimensional features $f_{vt}$ or $f_{it}$ can eventually be represented. Subsequent features are trained by fully connected layers, where each feature $f_{vt}$ or $f_{it}$ is viewed independently. The fully connected layer parameters designed for each person are shared, and each feature has its corresponding probability prediction output $p_i$ through the fully connected layer, which is used to calculate the identity loss $l_{id}$.

### 3.2. Design of Loss Function

The loss is mainly considered in the following two aspects: (1) cross-modality constraint, for the huge inter-modality differences, the core idea is that the distance of different IDs in different modalities should be greater than the distance of the same ID in different modalities, and the distance of different IDs in the same modality is greater than the distance of the same ID in different modalities; (2) intra-modality constraint, i.e., identity classification loss, to distinguish different samples in the same modality.

For the feature differences between same-modality and cross-modality, the loss function is designed as follows:

As shown in Figure 2, where the same color represents the same modality, the same border shape represents the same ID, and the cross-modality constraint target is that the distance of different IDs within the same modality is greater than the distance of the same ID within the cross-modality, i.e., $d(v_i, t_i) < d(v_i, v_j), d(v_i, t_i) < d(t_i, t_j)$. Similarly, the distance across different IDs of a modality should be greater than the distance across the same ID of a modality, i.e., $d(v_i, t_i) < d(v_i, t_j)$. The intra-modality constraint objective is to distinguish different IDs under the same modality.

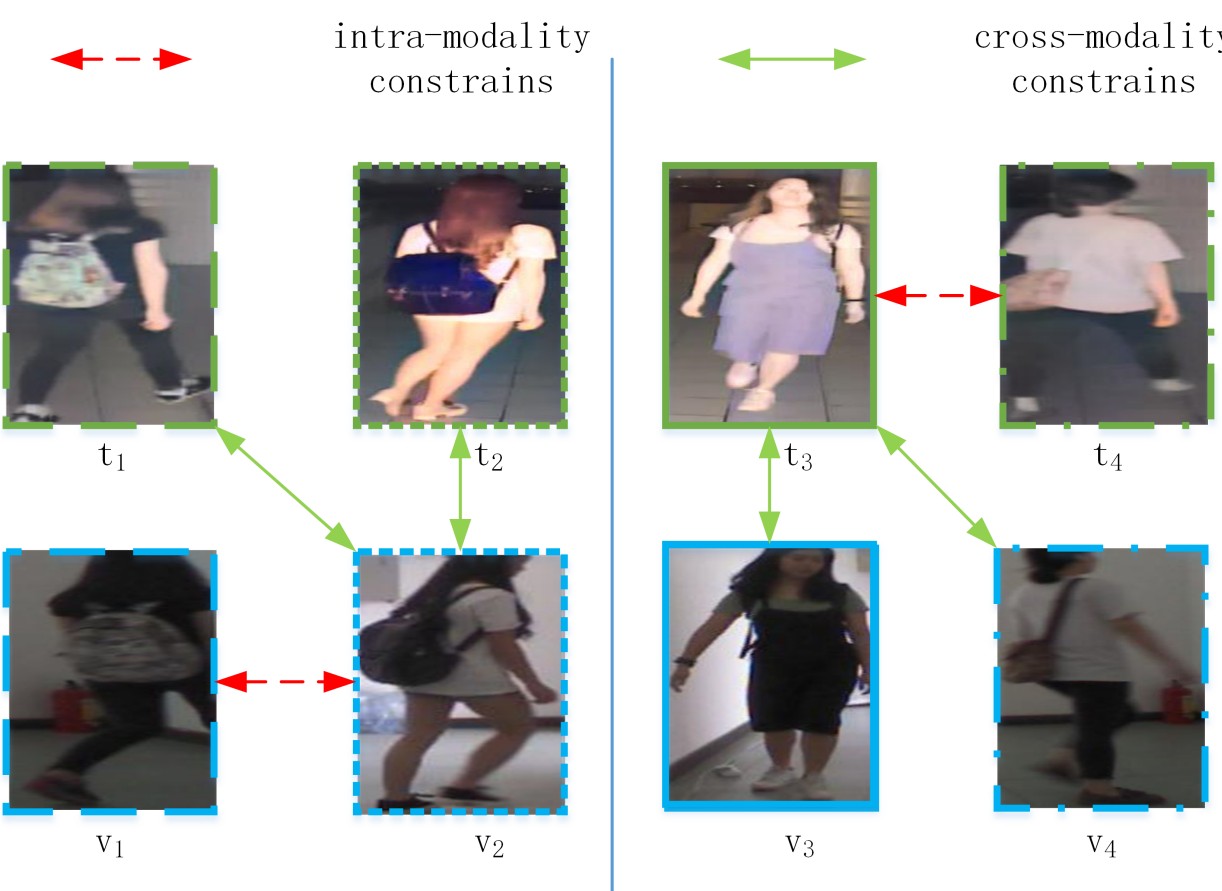

**Figure 2.** Design of the loss function.

In terms of objective function construction, a multi-objective joint optimization is performed as shown in Equation (1), including identity loss $l_{id}$ for sample identity identification under the same modality and cross-modality constraint $l_{cross}$.

$$l_{total} = l_{id} + l_{cross} \tag{1}$$

where $l_{id} = -\frac{1}{n} \sum_{i=1}^{n} log(p(y_i|x_i))$, n denotes the number of samples trained by each batch, and $p(y_i|x_i)$ denotes the predicted probability that the input image $x_i$ and its class label $y_i$, after softmax classification, $x_i$ is recognized as class $y_i$. $l_{cross}$ in turn includes the constraint $l_{cross1}$ between the distance of the same ID across modalities and the distance of different IDs within the same modality and the distance of different IDs across modalities and the distance of the same ID across modalities $l_{cross2}$.

$$l_{cross1} = \sum_{i=1}^{n} max[(p + d(v_i, t_i) - mind(v_i, v_j)), 0] \\ + \sum_{i=1}^{n} max[(p + d(t_i, v_i) - mind(t_i, t_j)), 0] \tag{2}$$

$$l_{cross2} = \sum_{i=1}^{n} max[(p + d(v_i, t_i) - mind(v_i, t_j)), 0] \\ + \sum_{i=1}^{n} max[(p + d(t_i, v_i) - mind(t_i, v_j)), 0] \tag{3}$$

where $v$ denotes visible data, $t$ denotes infrared data, and $i$, $j$ denote sample IDs, $p$ is a predefined threshold value.

## 4. Experiments

### 4.1. Dataset and Evaluation Criteria

**SYSU-MM01.** SYSU-MM01 is the first standard dataset in the field of cross-modality (RGB-IR) person re-identification, which consists of 6 cameras (4 RGB and 2 IR). The dataset contains 491 people with a total of 287,628 RGB images and 15,792 IR images. Each person is captured by at least two cameras with different positions and viewpoints. The training set consists of 395 people with 22,258 RGB images and 11,909 IR images. The test set consists of 96 people, with 3803 IR images as the query set and 301 randomly selected RGB images as the gallery set.

**RegDB.** RegDB [38] contains 412 different person identities; 10 RGB images and 10 thermal images are collected for each identity, and the weather conditions and the camera shooting viewpoint are the same in images of the same identity. There were 254 females and 158 males, and 156 out of 412 were photographed from the front and 256 from the back. The 4120 identity images of 206 individuals are selected as the training set, and the 4120 identity images of the remaining 206 individuals are used for testing. RegDB has two evaluation modes. One is visual-infrared matching, which searches for infrared images based on a given visible image. The other is infrared-visible matching, which uses a given infrared image to search for a visible image.

The recognition rates of rank1, rank10, and rank20 in the Cumulative Match Characteristic (CMC) curve were used as evaluation metrics. Higher values of all three indicate more accurate recognition in different settings. In addition, mean Average Precision (mAP), a commonly used evaluation metric in the field of information retrieval, was used as an evaluation method for the cross-modality person re-identification task. Higher values of mAP indicate better retrieval capabilities of the model.

### 4.2. Experimental Details

The model in this study is trained on Pytorch and implemented based on an NVIDIA 3080 GPU. Stochastic gradient descent (SGD) [39] is used for network optimization, and the initial learning rate is set to 0.01 and gradually decreases after every 30 iterations. The number of iterations (epoch) is set to 60, and each batch is sampled for 8 different identity data types, including 4 visible and 4 infrared data types; the batch size is set to 64 during training.

### 4.3. Experiments Result

**Comparison with existing methods.** To verify the effectiveness of the proposed algorithm, experiments are carried out on the SYSU-MM01 dataset in all search modes and indoor search modes according to the evaluation metrics in this section. The compared algorithms include traditional LOMO [40], HOG [41], basic algorithms such as One-stream, Two-stream, and zero-padding [23] based on feature learning, BDTR [42], D-HSMER [25] methods that add metric learning on top of representation learning, and GAN networks using the AlignGAN [27] algorithm, etc.

(1) All search mode

The cross-modality person re-identification algorithm based on a two-branch network proposed in this section achieves an accuracy of 52.3% for its rank-1 and 50.25% for mAP in the global search mode, which is a big improvement compared with the existing methods. Several representative VI-ReID algorithms are selected for comparison with the proposed method, the experimental results are presented in Table 2, and the performance comparison is presented in Figure 3.

(2) Indoor search mode

The proposed cross-modality person re-identification algorithm based on a two-branch network also performs well in indoor search mode, and its rank-1 accuracy reaches 59.25% accuracy and mAP reaches 63.74% accuracy in indoor search mode, which is a big improvement compared with the existing methods. The specific experimental results are shown in Table 3, and the performance comparison is shown in Figure 4.

As can be seen from Figure 5, the result of "indoor search mode" is more obvious than that of "all search mode", mainly because the attention information of indoor images is relatively uniform and the indoor background is more single than that of outdoor. The gap between the generated image and the actual image in the process of modal conversion is smaller, so the subsequent recognition effect will be better.

**Table 2.** VI-ReID results of different methods on SYSU-MM01 in all-search mode.

| Methods | r = 1 | r = 10 | r = 20 | mAP |
|---|---|---|---|---|
| HOG | 2.76 | 18.25 | 31.91 | 4.24 |
| MLBP | 2.12 | 16.23 | 28.32 | 3.86 |
| LOMO | 1.75 | 14.14 | 26.63 | 3.48 |
| GSM | 5.29 | 33.71 | 52.95 | 8.00 |
| One-stream | 12.04 | 49.68 | 66.74 | 13.67 |
| Two-stream | 11.65 | 47.99 | 65.50 | 12.85 |
| Zero-padding | 14.80 | 54.12 | 71.33 | 15.95 |
| CmGAN | 26.97 | 67.51 | 80.56 | 27.80 |
| BCTR | 16.12 | 54.90 | 71.47 | 19.15 |
| BDTR | 17.01 | 55.43 | 71.96 | 19.66 |
| AlignGAN | 42.4 | 85.0 | 93.7 | 40.7 |
| Hi-CMD | 34.9 | 77.6 | - | 35.9 |
| AGW | 47.5 | 84.39 | 92.14 | 47.65 |
| VI-TBNDC(ours) | **52.30** | **88.6** | **95.5** | **50.25** |

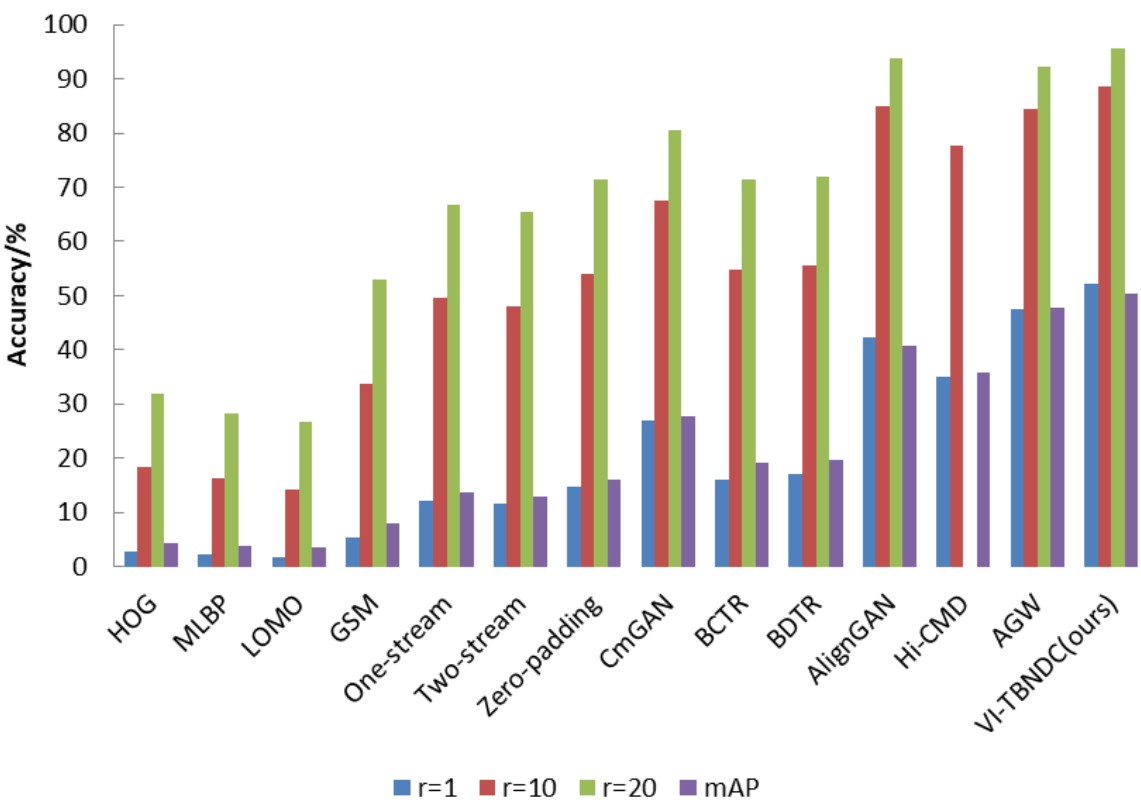

**Figure 3.** Performance comparison of different methods on SYSU-MM01 in all-search mode.

**Table 3.** VI-ReID results of different methods on SYSU-MM01 in indoor-search mode.

| Methods | r = 1 | r = 10 | r = 20 | mAP |
|---|---|---|---|---|
| Zero-padding | 20.58 | 68.38 | 85.79 | 26.92 |
| TONE | 20.82 | 68.86 | 84.46 | 26.38 |
| HCML | 24.52 | 73.25 | 86.73 | 30.08 |
| cmGAN | 31.63 | 77.23 | 89.18 | 42.19 |
| eBDTR | 32.46 | 77.42 | 89.62 | 42.46 |
| MAC | 36.43 | 62.36 | 71.63 | 37.03 |
| MSR | 39.64 | 89.29 | 97.66 | 50.88 |
| AlignGAN | 45.9 | 87.6 | 94.4 | 54.3 |
| AGW | 54.17 | 91.14 | 95.8 | 62.97 |
| VI-TBNDC(ours) | **59.25** | **91.32** | **97.64** | **63.74** |

Among the compared methods, LOMO [40] and HOG [41] apply traditional algorithms that cannot extract useful features at this stage, so the accuracy is low. One-stream, Two-stream, and zero-padding [23] are cross-modality person re-identification methods based on feature learning, which only perform representation learning without further metric learning, so the results are not too good. BDTR [42] and D-HSMER [25] add metric learning to the feature learning, so the recognition accuracy is higher than the accuracy using only feature learning. Although BDTR [42] also uses a two-branch network for different modality feature extraction, its backbone network, AlexNet [43], has low depth and cannot extract more feature information, which results in poor performance. AlignGAN [27] is based on a generative adversarial network with modality transformation, converting visible (or IR) images into IR (or visible) modality images, narrowing the modality differences, and unifying the images to the same modality; thus, the recognition accuracy will be better. The proposed method builds discriminative features based on the reduction of inter-modality

differences while using the dual constraints of the two-branch network, which leads to better performance on the SYSU-MM01 dataset.

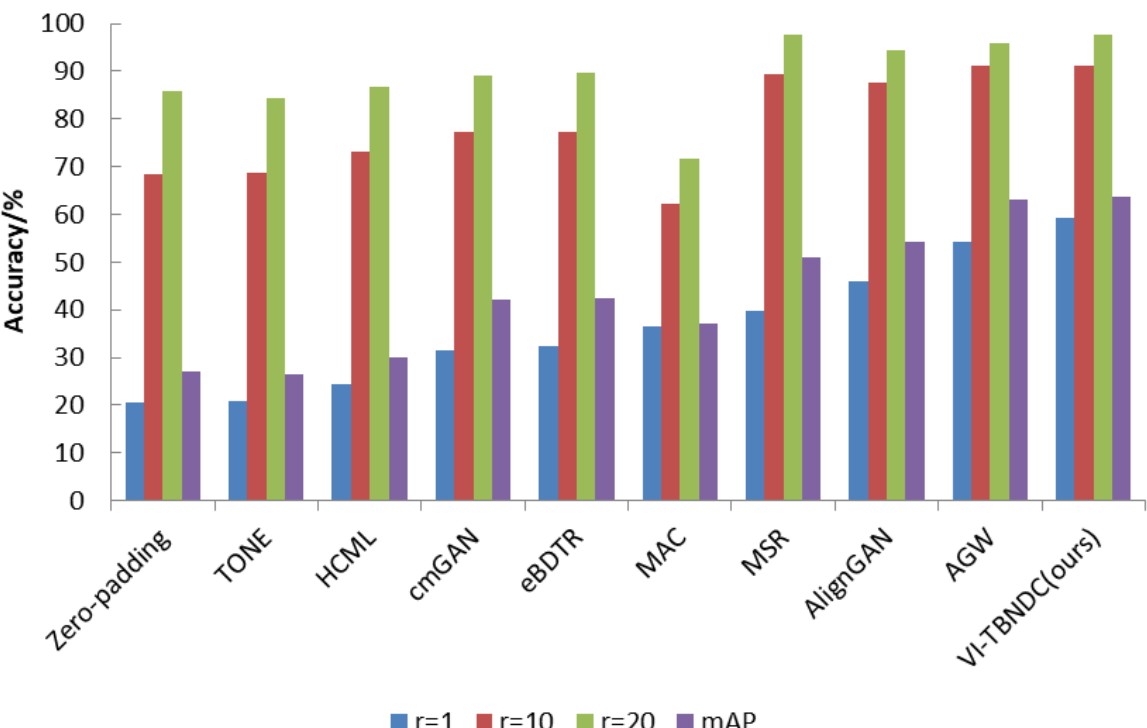

**Figure 4.** Performance comparison of different methods on SYSU-MM01 in indoor-search mode.

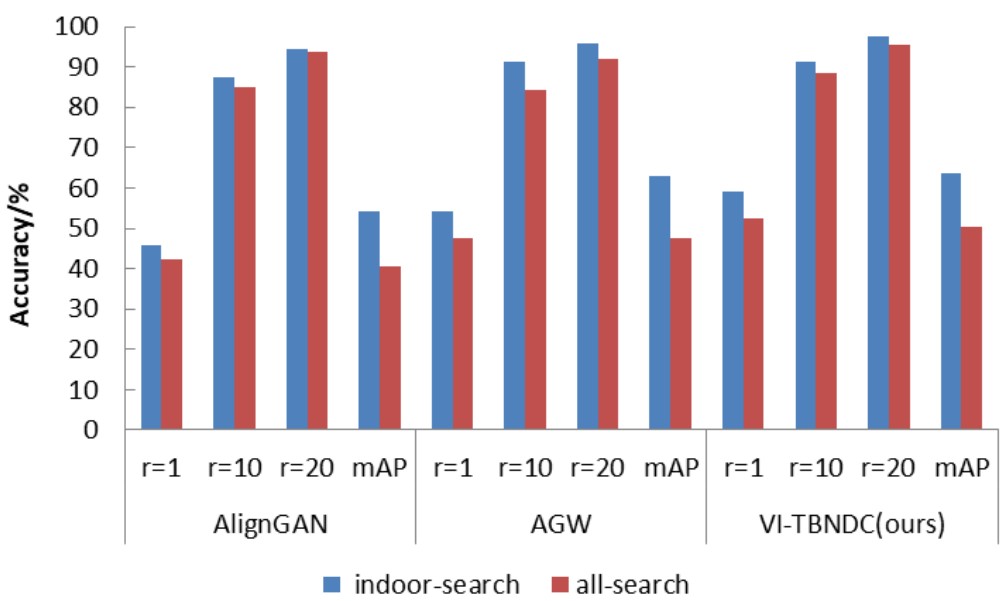

**Figure 5.** Comparison of indoor search and all search results.

Since infrared and visible images are taken with different modalities, they have drastically different appearances. Therefore, it is not efficient to map them directly to the feature space. To alleviate this problem, this paper uses GAN to colorize infrared images to reduce the modality discrepancy by unifying the image representation. In addition, color images can also be grayscaled to reduce the modality discrepancy.

The experimental results are presented in Table 4, where the differences between the two modality conversion methods are mainly reflected in rank-1 and mAP scores. Among them, in the all-search mode, infrared image colorization has 1.16% and 2.60% higher rank-1 and mAP scores than color image grayscale, respectively. In the indoor search mode, infrared image colorization has 4.36% higher rank-1 and 1.62% higher mAP scores than color image grayscale. Infrared images have no color information, and although graying color images reduces their color difference, NIR images have much weaker dependence on R, G, and B than they do on each other, resulting in NIR images that differ from grayscale images converted from RGB images [19].

**Table 4.** Results of different modality conversion methods on SYSU-MM01.

| Methods | All-Search | | | | Indoor-Search | | | |
|---|---|---|---|---|---|---|---|---|
| | r = 1 | r = 10 | r = 20 | mAP | r = 1 | r = 10 | r = 20 | mAP |
| VI-TBNDC(gray) | 51.14 | 88.39 | 95.21 | 47.65 | 54.89 | 91.66 | 97.17 | 62.12 |
| VI-TBNDC(ours) | **52.30** | **88.60** | **95.50** | **50.25** | **59.25** | **91.32** | **97.64** | **63.74** |

### 4.4. Ablation Study

**Effectiveness of pre-processing the IR image with colorization.** The infrared data is converted into color images by using the generative adversarial network, and the infrared images with only one channel are converted into RGB images with three channels, which reduces the difference between modalities to a certain extent and provides great convenience for cross-modality person re-identification.

The specific conversion process consists of two stages, an image generation stage and an image discrimination stage. First, in the image generation stage, the model learns to colorize a given input such that the original infrared image becomes an RGB image. The specific structure of the generator G is shown in Figure 6.

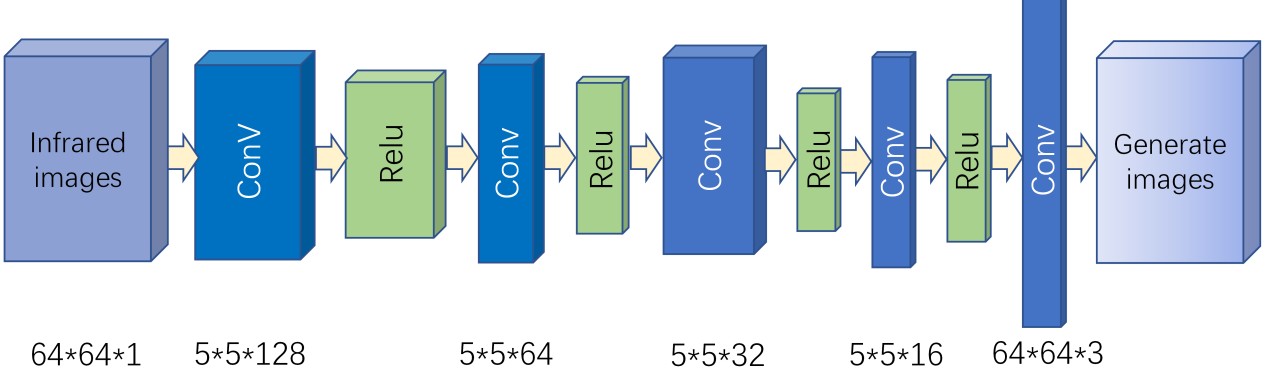

**Figure 6.** Structure diagram of the generator G.

In the second stage, the discriminative model is used to estimate the probability that the generated image comes from the training dataset. The specific structure of the discriminator D is shown in Figure 7. The generative model G is obtained by training from NIR images to produce color RGB images. In addition, the discriminative model D is trained to assign the correct label to the generated color image according to the provided real color image, and finally, the purpose of modal conversion is achieved.

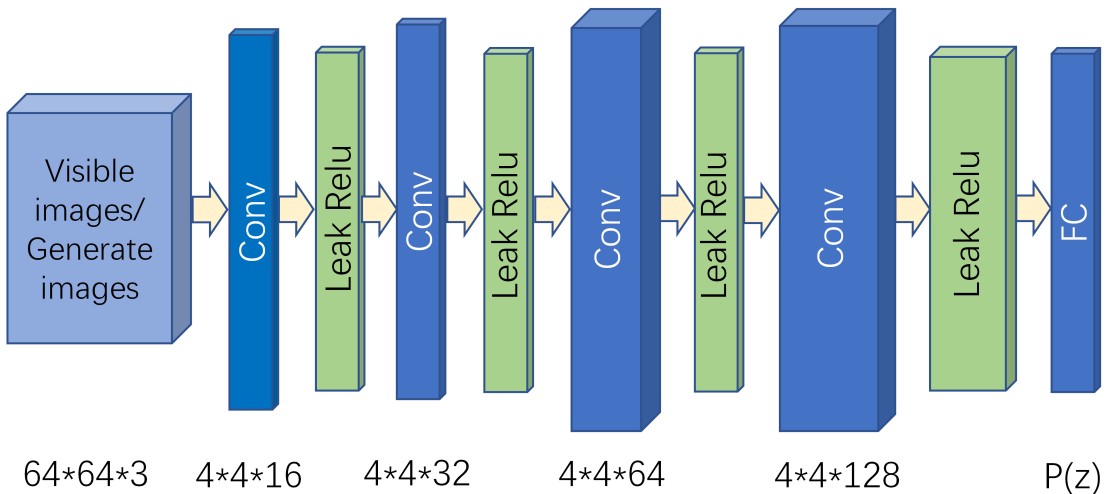

**Figure 7.** Structure diagram of discriminator D.

The generator (G) and discriminator (D) are both feed-forward neural networks that play a min-max game against each other. The generator converts near-infrared images to visible images. A set of data (real image (z) or a generated image (G(z))) is fed into the discriminator to generate the probability that this data is real (P(z)). The discriminator is optimized to improve its probability for real data (given real images) and reduce its probability for fake generated data (miscolored NIR images), as introduced in reference [44], and the specific process is shown in Equation (4).

$$\triangledown \theta_g \frac{1}{m} \left[ \log D\left(x^i\right) + \log\left(1 - D\left(G\left(z^{(i)}\right)\right)\right) \right] \tag{4}$$

where m is the number of samples in each batch, x is the real image, and z is the color NIR image generated by the network. The weights of the discriminator network (D) are updated by boosting its stochastic gradient.

On the other hand, in order to increase the probability that the generated data will be highly rated, the generator will be optimized, and the specific process is shown in Equation (5).

$$\triangledown \theta_g \frac{1}{m} \sum_{i=1}^{m} \log 1 - D\left(G\left(z^{(i)}\right)\right) \tag{5}$$

where m is the number of samples in each batch, and z is the color NIR image generated by the network. As in the previous case, the weights of the generator network (G) are updated by decreasing its stochastic gradient. Figure 8 shows part of the infrared image colorization effect.

As can be seen from Table 5, the rank1 metric improves by 2.7% and the mAP metric improves by 3.8% compared with the baseline on the SYSU-MM01 dataset after preprocessing the infrared images with colorization, which indicates that colorization of the infrared images does have the effect of reducing the inter-modality differences, and by this operation the infrared images can be converted to visible images. The reduction of modality differences between the two provides convenience for feature extraction as well as feature matching in the subsequent person re-identification network, and the improvement in metrics verifies the effectiveness of the conversion network in cross-modality person re-identification.

**Table 5.** Comparison of infrared image colorization pre-processing experiments.

| Pre-Processed or Not | r = 1 | r = 10 | r = 20 | mAP |
|---|---|---|---|---|
| No pre-processing | 49.8 | 85.2 | 92.2 | 48.60 |
| Pre-processed | 51.3 | 88.6 | 95.5 | 50.25 |

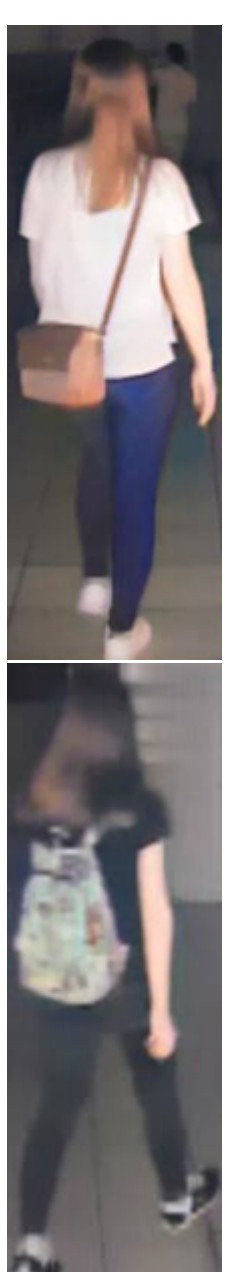

**Figure 8.** Infrared image colorization results.

**Effectiveness of backbone.** Ye et al. in the literature [42] also mentioned the use of a two-branch network for feature information extraction of two modalities, where the backbone network used is AlexNet [43] and the backbone network used in this paper is ResNet50. Here, different backbone networks are used to perform feature extraction on the SYSU-MM01 dataset to validate the rationality of the chosen backbone network.

As can be seen from Table 6, the best results were achieved with ResNet50 as the backbone network of the two-branch cross-modality person re-identification algorithm. Due to the different depths of the networks, there are differences in the extracted image

features, and thus the model performance can also vary significantly. Specifically, the AlexNet network was proposed earlier, and the various capabilities are weaker compared with the ResNet network. AlexNet consists of an 8-layer structure, of which the first 5 layers are convolutional layers and the next 3 layers are fully connected layers, and the features extracted in the convolution process are limited, so it cannot achieve better results. The mAP can only reach 20.3%, and the performance of rank-1 is only 20.5%. ResNet18 has increased the number of layers compared with AlexNet and added the residual network structure, so the effect is significantly better than AlexNet, and the speed is relatively faster due to the fewer layers of the network, but the deeper pedestrian features cannot be extracted, so the performance is poor in the ResNet series backbone network. ResNet34 has more layers than ResNet18 and has improved feature extraction ability; thus, the performance has improved, with 3.2% and 1.9% improvements in mAP and rank1 metrics, respectively, but it is still not the best. ResNet50 has the best performance among the three, with 50.25% in the mAP metric and 52.30% in rank1 metric.

**Table 6.** Performance comparison of different backbone networks.

| Method | Backbone | mAP | Rank-1 |
|---|---|---|---|
| VI-TBNDC | AlexNet | 20.3 | 20.5 |
| | ResNet18 | 43.6 | 45.3 |
| | ResNet34 | 46.8 | 47.2 |
| | ResNet50 | 50.25 | 52.30 |

*4.5. Visualization of the Results*

Figure 9 shows the visualization of the proposed method in this paper; three samples are selected and the top 10 images closest to them are displayed; the results are correctly indicated with green borders, and red means not the same id as the sample; the results show that the proposed method in this paper has better performance on the SYSU-MM01 dataset.

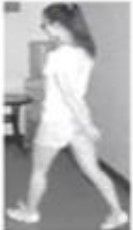

target 1 to be queried

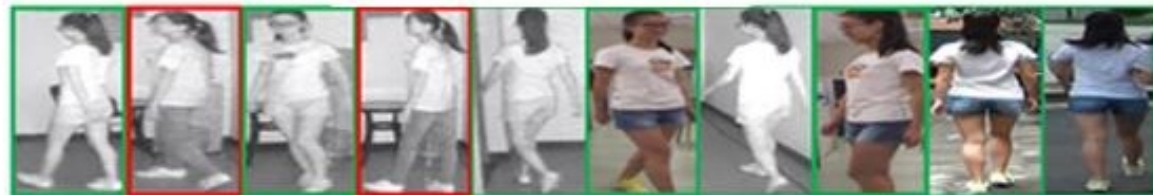

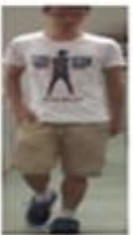

target 2 to be queried

**Figure 9.** *Cont.*

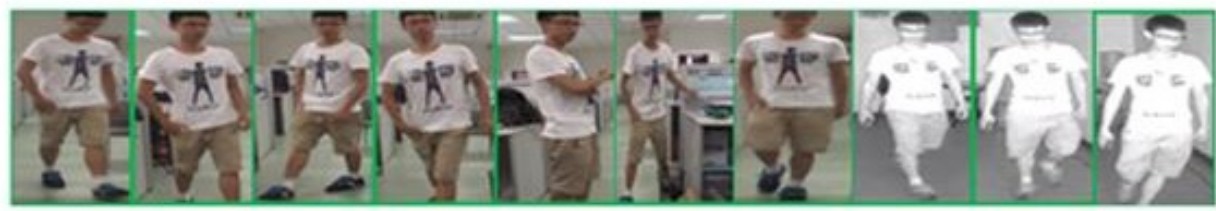

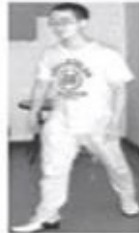

target 3 to be queried

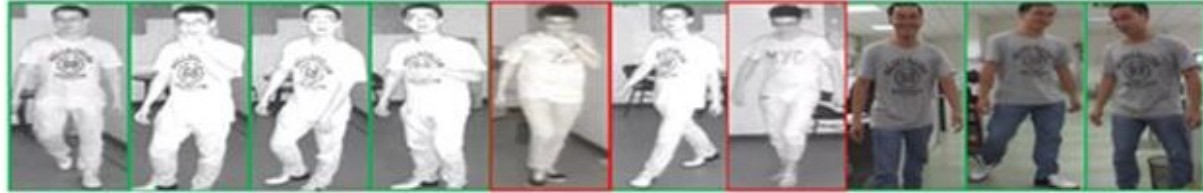

**Figure 9.** Visualization of the results.

## 5. Conclusions

This paper proposes a cross-modality person re-identification method based on a two-branch network for infrared and visible images. Firstly, the method introduces the infrared image colorization technique into the field of person re-identification and uses this technique to reduce inter-modality differences and achieve better results. Furthermore, a two-branch network is proposed, using ResNet50 as the backbone of the feature extraction network and extracting feature information common to both modality data and unique to each modality separately. Using a parameter-independent network structure in the shallow layer and a weight-sharing network structure after the convolutional layer conv2. The low-level semantic features of the two modalities are extracted separately, and then the two modalities features are embedded in the same network to extract the high-level semantic features with identity discriminative information. Finally, a cross-modality hybrid loss function is proposed for the feature differences between same-modality and cross-modality, which effectively corrects the model using the mutual constraints between the modalities and effectively mitigates the negative impact of modality differences on cross-modality person re-identification. Good results are obtained in the SYSU-MM01 dataset using a near-infrared camera, demonstrating the usefulness of the proposed method in the cross-modality person re-identification task. During the experimental analysis, the performance of different backbone networks under the same dataset is verified, justifying the choice of ResNet50 as the backbone network. It also verifies the superiority of this network among current cross-modality person re-identification algorithms and analyzes the reasons for its better performance compared with other classical algorithms.

**Author Contributions:** Conceptualization, J.S. and J.Y.; methodology, J.S.; software, J.S.; validation, C.Z. and K.X.; formal analysis, J.S., J.Y. and K.X.; investigation, J.Y.; resources, J.Y.; data curation, J.S. and J.Y.; Writing—Original draft preparation, J.Y.; writing— Review and Editing, C.Z. and K.X.; visualization, J.Y.; supervision, J.S. and K.X.; project administration, J.S. and K.X.; funding acquisition, K.X. All authors have read and agreed to the published version of the manuscript.

**Funding:** This research was funded by the National Natural Science Foundations of China under grand No. 62272364.

**Institutional Review Board Statement:** Not applicable.

**Informed Consent Statement:** Not applicable.

**Data Availability Statement:** The data presented in this study are available on request from the corresponding author.

**Acknowledgments:** The authors would like to thank the Assistant Editor of this article and anonymous reviewers for their valuable suggestions and comments.

**Conflicts of Interest:** The authors declare no conflict of interest.

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
