# Peer review of "Cross-Modality Person Re-Identification Algorithm Based on Two-Branch Network"

_electronics, doi:10.3390/electronics12143193_

Round 1

Reviewer 1 Report

The authors of this article present a study on cross-modality (IR and color images) applied to the person re-identification in multi-view video sequences.

The methodology, which is clearly explained, relies on color encoding of IR images on the one hand, and on neural networks based on the Resnet-50 model on the other.

Experimental results show that the approach proposed by the authors achieves superior performance compared with the state of the art.

In order to improve the quality of this article, the following points need to be addressed:

1-    The choice to colorize IR images is certainly coherent. However, color encoding IR images generates color differences with respect to the ground truth. Another solution would be to do the opposite: retain the native IR images and transform the color images into grayscale. It would therefore be advisable to quantify the performance of such an approach and report it in a new version of the article.

2-    As indicated in the literature, the Resnet-50 backbone is not the best compromise in terms of classification performance vs. complexity. EfficientNet-type models are more efficient. A further study of the contribution of such models is welcome for the new version of the article.

Reviewer 2 Report

Author mentioned feature extraction of different modalities, without explanation how features are constructed, methodology of feature extraction. This is crucial element without detailed explanation and that should be improved. Especially that should be explained in light of two branch network, modality and identity feature , likewise feature embedding .

It would be interesting to have comparative study for inclusion of additional layer with fuzzy logic for better performance on top of proposed  model. 

Author mentioned feature extraction of different modalities, without explanation how features are constructed, methodology of feature extraction. This is crucial element without detailed explanation and that should be improved. Especially that should be explained in light of two branch network, modality and identity feature , likewise feature embedding .

It would be interesting to have comparative study for inclusion of additional layer with fuzzy logic for better performance on top of proposed  model.

Reviewer 3 Report

The paper proposed a two-branch cross-modality person re-identification algorithm and utilized infrared image colorization to pre-process IR images. Converting IR data into color data is novel and helps reduce the discrepancies between inter-modality images. The method is evaluated on public datasets and shows improvement compare to existing methods. The paper also presents comprehensive ablation studies to show the effectiveness of the design. The paper needs some polish and edits on languages.

Round 2

Reviewer 1 Report

The authors have answered my questions and made the corresponding corrections in the new version of their article.
I therefore suggest accepting this new version.